# Updates on Wound Infiltration Use for Postoperative Pain Management: A Narrative Review

**DOI:** 10.3390/jcm10204659

**Published:** 2021-10-11

**Authors:** Dusica M. Stamenkovic, Mihailo Bezmarevic, Suzana Bojic, Dragana Unic-Stojanovic, Dejan Stojkovic, Damjan Z. Slavkovic, Vladimir Bancevic, Nebojsa Maric, Menelaos Karanikolas

**Affiliations:** 1Military Medical Academy Medical Faculty, University of Defense, 11050 Belgrade, Serbia; bezmarevicm@gmail.com (M.B.); vladaban2004@yahoo.com (V.B.); maricvma@gmail.com (N.M.); 2Department of Anesthesiology and Intensive Care, Military Medical Academy, 11050 Belgrade, Serbia; 3Clinic for General Surgery, Military Medical Academy, 11050 Belgrade, Serbia; damjanslavkovic@gmail.com; 4University of Belgrade School of Medicine, 11050 Belgrade, Serbia; subojic@yahoo.com (S.B.); dragana.unic@gmail.com (D.U.-S.); 5Clinic for Anesthesia and Resuscitation, University Hospital Center “Dr Dragisa Misovic-Dedinje”, 11050 Belgrade, Serbia; 6Clinic for Anesthesia and Intensive Care, Cardiovascular Institute Dedinje, 11050 Belgrade, Serbia; 7Thoracic Surgery Unit, King Abdul Azis Specialist Hospital, Qurwa, Taif 26521, Saudi Arabia; dekonja45@yahoo.com; 8Urology Clinic, Military Medical Academy, 11050 Belgrade, Serbia; 9Clinic for Cardiothoracic Surgery, Military Medical Academy, 11050 Belgrade, Serbia; 10Department of Anesthesiology, Washington University School of Medicine, St. Louis, MO 63110, USA; menelaos.karanikolas@wustl.edu

**Keywords:** anesthetics, local/administration, dosage, catheters, indwelling, pain, postoperative therapy

## Abstract

Local anesthetic wound infiltration (WI) provides anesthesia for minor surgical procedures and improves postoperative analgesia as part of multimodal analgesia after general or regional anesthesia. Although pre-incisional block is preferable, in practice WI is usually done at the end of surgery. WI performed as a continuous modality reduces analgesics, prolongs the duration of analgesia, and enhances the patient’s mobilization in some cases. WI benefits are documented in open abdominal surgeries (Caesarean section, colorectal surgery, abdominal hysterectomy, herniorrhaphy), laparoscopic cholecystectomy, oncological breast surgeries, laminectomy, hallux valgus surgery, and radical prostatectomy. Surgical site infiltration requires knowledge of anatomy and the pain origin for a procedure, systematic extensive infiltration of local anesthetic in various tissue planes under direct visualization before wound closure or subcutaneously along the incision. Because the incidence of local anesthetic systemic toxicity is 11% after subcutaneous WI, appropriate local anesthetic dosing is crucial. The risk of wound infection is related to the infection incidence after each particular surgery. For WI to fully meet patient and physician expectations, mastery of the technique, patient education, appropriate local anesthetic dosing and management of the surgical wound with “aseptic, non-touch” technique are needed.

## 1. Introduction

In the past decade we have witnessed a significant shift towards regional analgesia as the primary technique in postoperative pain management. Single wound infiltration with local anesthetic (WI) or continuous local anesthetic infusion through catheters placed into the surgical wound (continuous wound infiltration, CWI) have recently been re-introduced as integral parts of multimodal analgesia schemes for postoperative pain control following various surgical procedures under general or regional anesthesia [1]. Wound infiltration (WI) with local anesthetics (LA) is used as the main anesthetic for minor surgeries, such as repair of lacerations, skin surgery and treatment of painful oral or genital lesions, but can also be used as supplement to general anesthesia in several types of surgical procedures. CWI improves postoperative analgesia quality and shows an opioid-sparing effect [2]. The term “local infiltration analgesia” (LIA) is used to describe the application of “high volume of diluted, long-acting local anesthetic” in tissue structures (usually in knee or hip surgery) to provide analgesia and is the term we use in this manuscript [3,4,5].

Advantages of recommended WI techniques are safety, simplicity, and enhanced postoperative analgesia, especially during mobilization [6,7]. In recent years, procedure-specific postoperative pain (PROSPECT) Working Group recommended WI for open abdominal surgeries (Cesarean section, colorectal surgery, abdominal hysterectomy, herniorrhaphy), laparoscopic cholecystectomy, oncological breast surgeries, laminectomy, hallux valgus surgery and radical prostatectomy (Figure 1) [8,9]. Although CWI confers several benefits, including improved analgesia, reduced opioid use, reduced side effects, increased patient satisfaction and reduced length of hospital stay (LOS), CWI use is limited because of concerns about wound catheter displacement, infection risk, cost and misjudgment of the technique [10].

Published reviews and books discuss WI techniques and appropriate use of local anesthetics [6,11]. Currently, WI is used cautiously due to concerns about wound infection. Furthermore, WI is not always used in accordance with the recommended technique and local anesthetic safety. The aim of this review is to update our knowledge on the application of WI techniques in surgical practice and inspire its use as a step in multimodal pain management.

## 2. Materials and Methods

This narrative review is a focused evaluation of published data on the utilization of single-shot WI or CWI in adult patients for management of postoperative pain. Therefore, we did not include international criteria/PRISMA and we did not grade studies by level of evidence.

We searched PubMed for abstracts in English, using the terms “wound infiltration AND postoperative pain AND”, “neurosurgery”, “cardiac surgery”, “trauma surgery”, “emergency cases”, “thoracic surgery”, “abdominal surgery”, “breast surgery”, “thyroid surgery”, “day case surgery”, “urology surgery”, “gynecology surgery”, “othopedics”, “wound infection”, “wound bleeding” published in the past 20 years (the date of literature search was 12 December 2020). Because use of WI in plastic surgery is discussed in great detail in the literature, we excluded it from this review. Based on expert opinion, we included open and blinded studies, reviews and meta-analysis, and available commentaries and editorials related to the MESH terms.

Because children and their parents/family have additional factors to consider, WI in children was not part of this review. The references used in this publication are chosen from the published materials and encourage further exploration of the topic.

## 3. Wound Infiltration Technique

Surgical site infiltration can be used as sole anesthetic for minor superficial surgical procedures, administered in the subdermal and musculofascial planes, or instilled in a cavity (e.g., intra-articular administration for joint surgery or intraperitoneal administration for abdominal surgery) [2,12,13,14]. Infection at the site of injection, the true LA allergy and patient refusal are the only contraindication for WI [15]. Surgical site infiltration requires knowledge of anatomy and the source of pain from surgical procedure and includes systematic and extensive infiltration of LA in various tissue planes under direct visualization before surgical wound closure or preoperatively at the planned incision line. Explanation to the patient about the feeling of touch and pull of tissue when surgery is performed under WI is necessary [16].

Use of smaller diameter needles (27- to 30-gauge) [17,18] slows injection rate and consequently reduces pain during injection [18,19]. Intradermal injection of LA can induce anesthesia more rapidly than subcutaneous injection [20]. The anatomy of nerve endings localized in the dermis resembles leaves on a tree with larger branches and trunks in the fat. Intradermal injection produces more pain by stretching dense nociceptor-rich dermal tissue, rather than by stretching of loose subcutaneous tissue [19]. For intradermal injection, needle insertion at a 90 degree angle during skin penetration passes through fewer pain fibers and reduces pain (Figure 2) [21]. The pain can be minimized by injecting just below the dermis into the subcutaneous fat at the nerve trunk level just beneath branching. The presence of immediate whitening, tightening, or “peau d’orange” appearance of the skin are signs of intradermal injection [22]. Skin nociceptors respond to rapid tissue distension and stretching [20]; therefore, slow injection facilitates “accommodation” of nerve endings and provides time for LA diffusion and blockade of impulse transduction in stimulated fibers [19]. Injection of 0.2 to 0.5 mL of LA immediately following needle insertion, followed by a pause, is associated with reduced pain compared to continuous injection of 2 mL or higher volume from the beginning by enabling the LA to numb the needle insertion site [19,21]. With good technique, it is possible to anesthetize extensive areas, while the patient only feels pain at the initial puncture. After perpendicular insertion of the needle for the initial LA injection it is possible to proceed with intradermal insertion of the needle obliquely and tangentially. The clinician performing the block needs to reinsert the needle within 1 cm of blanched anesthetized skin, while the other hand palpates the extent of the tumescence.

Proper planning of local anesthetic injection is important for optimizing analgesia [23]. WI in a superficial plane is less helpful compared to infiltration between the muscle layer and peritoneum in abdominal surgery [24,25], because somatic pain originates from deep layers of the abdominal wall, including the peritoneum [26]. Frequency of LA administration also influences postoperative analgesia and CWI is superior compared to bolus or intermittent LA administration [26,27,28] with regards to meaningful reduction in opioid use and improved pain outcomes [29]. A recent meta-analysis showed that CWI through preperitoneal catheters is more effective compared to CWI through subcutaneous catheters, and can be used as alternative to epidural analgesia after abdominal surgery [30].

WI at the end of surgery includes infiltration of all layers involved in the surgical incision and throughout the wound length [10]. For abdominal surgical procedures, infiltration should be performed in the peritoneal, musculofascial, and subdermal tissues, whereas infiltration of the capsule and other soft tissues is performed in major joint surgery. The injecting needle for surgical site infiltration should be different than for standard local anesthetic administration as described previously. Surgical site infiltration is ideally performed with a short 22-gauge needle with direct visualization of tissue layers [2]. Recently published studies precisely describe the infiltration process and can serve as practical manuals. Although pre-incisional block is theoretically preferable [6] it is rarely studied [31,32,33,34,35] and WI is usually done at the end of surgery [10].

Compared to WI, CWI provides prolonged analgesia using different delivery modes, including patient-controlled analgesia (PCA), continuous infusion, or intermittent bolus [10]. However, implementation of CWI techniques is more complex, because it requires devices such as catheters, an elastomeric pump for continuous delivery of LA, surgical experience with the technique and education of patients and medical staff [10,36]. CWI effectiveness is higher in skin regions with dense subcutaneous and connective tissue, compared to areas with variable innervation [36].

The advisory on local anesthetic systemic toxicity (LAST) prevention is to use incremental LA injections with administration of small doses (up to 5 mL) after aspiration and time between LA injections should be up to 30 s, ideally one circulation time (30–45 s) [37,38]. Larger dose increments need longer time intervals between injections to reduce LA accumulation [37]. CWI increases analgesia quality and has opioid-sparing effect; therefore, it is recommended when other techniques are not available [36].

## 4. Local Anesthetics and Medications for Wound Infiltration

Local anesthetics block voltage-gated sodium channels of nerve endings [39]. Application of LA directly to wounds provides analgesia by blocking the transmission of pain signals from nociceptive afferents in the wound surface and by inhibiting local inflammatory response to injury, thereby reducing the release of inflammatory mediators from neutrophils and decreasing edema formation [10]. Local anesthetics have pleiotropic effects, such as antioxidant, anti-hyperalgesic, and neuroprotective properties [40]. A major limitation of the WI technique is the limited duration of the LA analgesic effect; this limitation can be addressed with insertion of specially designed multi-orifice catheters for CWI.

Lidocaine alone or in combination with epinephrine is the most widely used LA since its introduction in 1948 [41]. Lidocaine has high tissue permeability and diffuses rapidly from the skin to adjacent tissues. Benefits of adding epinephrine to lidocaine include reduced bleeding, prolonged action and higher maximum allowed dose due to slower vascular diffusion. Furthermore, epinephrine causes a blanching effect on the skin, thereby providing a helpful visual indicator of the anesthetized area’s extent [19,42]. Lidocaine (1%) with 1:100,000 epinephrine has a pH of 4.2, which is more acidic than physiological pH and causes greater pain intensity during injection [43].

In contrast, buffering lidocaine with 10:1 sodium bicarbonate is associated with significantly reduced pain on injection [19,43]. Lidocaine buffering is done by drawing 1 mL of 8.4% sodium bicarbonate and filling the syringe with 9 mL of lidocaine, in order to achieve the desired 10:1 ratio. LA solutions like bupivacaine, ropivacaine, or liposomal bupivacaine are used more often for surgical site infiltration (WI and CWI) than for local anesthesia of intact skin.

The maximum LA dose is determined in mg based on patient weight and risk factors; and the allowable amount can be diluted with preservative-free normal saline to the total volume needed, based on incision size [2]. Warming the local anesthetic significantly reduces pain on injection whether the solution is buffered or not [44], presumably because cold temperature stimulates more nociceptor fibers whereas increased temperature accelerates diffusion of local anesthetic molecules across cell membranes, producing a quicker onset of effect [19,44].

Authors consider the central nervous system toxicity and cardiotoxicity of bupivacaine and the untoward effects of accidental intravascular injection or systemic absorption of epinephrine, so they tend to use local anesthetic combinations. Combination of 1% lidocaine with 0.5% bupivacaine for WI has advantages, as lidocaine provides rapid onset of effect and dense sensorimotor block, while bupivacaine prolongs the anesthetic effect. Adding epinephrine extends anesthesia duration and motor blockade, but when discussing WI, motor block is not a pertinent consideration [16,45]. Ropivacaine is a long-acting amino amide local anesthetic, with decreased neurotoxicity and cardiotoxicity potential [46].Ropivacaine (0.25–0.75%) decreases regional blood flow at the injection site via peripheral vasoconstriction effects. The mixture of ropivacaine with lidocaine combines the rapid onset of lidocaine with the long duration of ropivacaine.

Other medications used off-label for CWI, alone or as adjuncts to LA include alpha-2-agonists [47,48], tramadol [49], nonsteroidal anti-inflammatory drugs (NSAIDs) [50,51,52,53] and NMDA antagonist [54]. Of note, studies using adjuvants rarely compare the adjuvant given intravenously in the same dose in order to assess the systemic vs. local effect of the adjuvant [48,50]. Furthermore, studies rarely report side effects related to the use of adjuvant medications [48].

Whenever WI is combined with regional anesthesia, it is important to carefully calculate the total safe dose of LA in order to reduce the risk of toxicity. The incidence of local anesthetic systemic toxicity after subcutaneous infiltration is 11%, and other resources discuss more on this topic [55]. It is vital to limit the LA dose based on patient ideal body weight (IBW) [56] and risk factors (age, lower muscle mass, lower ejection fraction, liver and renal insufficiency, and metabolic disorders) [55]. Intralipid availability is mandatory for immediate use “at the first signs” of LAST, together with resuscitation equipment and benzodiazepines [37]. Although bupivacaine has higher potential for cardiac toxicity compared to lidocaine, lidocaine is more frequently involved in LAST [55]. In our practice, surgeons are reluctant to give lidocaine without previous dosage calculation, especially to top up regional blocks with WI. All agents are safe if we respect recommended dosing “using the lowest concentration and dose necessary for analgesia” (Table 1) [37,55,57,58,59,60,61].

The rate of systemic LA absorption differs between injection sites due to vascularity of the area and tissue binding of LA [62]. Further research is needed for development of specific LA dosing guidelines related to surgical site for single WI [63]. The study group led by Borgeat and Rawal summarized findings from literature and presented regimes for CWI [4] based on GRADE system for quality of evidence assessment [64]. In our review we adapted their table and summarized the recommendations (Table 2) [4,65,66,67,68,69,70,71,72,73,74].

Applications available on electronic devices (like SafeLocal, Johns Hopkins University) can help with safe LA dosing [75]. Because LA toxicity is additive [37,76] and there is no clear dosage recommendation for mixing local anesthetics for WI [60], use of the lowest effective dose, aspiration before LA injection, and use of incremental injections are reasonable LAST preventive measures [37,55].

## 5. Complications of Wound Infiltration

An animal study showed that pre-incisional WI with bupivacaine and ropivacaine did not adversely impact histological wound healing and did not reduce tensile strength of the infiltrated tissue [77]. Local anesthetics have a broad spectrum of effects, including antimicrobial, anti-inflammatory, and even pro-inflammatory properties that may affect wound healing [78,79]. LAs cause vasodilatation by direct action on vascular smooth muscle. Furthermore, LAs decrease peripheral vasoconstriction, thus preventing reduction of perfusion and oxygenation of tissue surrounding the surgical wound [80].

Complications after WI are rare, but can include local anesthetic toxicity [81], wound infection [81,82], hematoma [81], and bruising [83]. Wound infection is particularly troubling: concern about infection is probably one of the main obstacles for WI, but published data show low infection risk both in active (0.7%) and control groups (1.2%) [10]. Unintentional puncture of superficial vessels during WI can cause superficial bruising or hematoma. Bruising resolves spontaneously, but it is important to inform patients about this risk [83]. Surgeons use drainage and prophylactic antibiotic therapy in hematoma treatment [81].

CWI requires additional time at the end of surgery and involves placement of special catheters, thereby increasing cost and raising concern about infection risk. Reported CWI complications include wound infection, catheter leakage, kinking or obstruction, failure to infuse due to obstruction, unintentional removal and, inappropriate tubing management [84]. The incidence of wound infection ranges from 1.2% overall to 13.8% for hepatobiliary surgery, and it does not exceed the baseline incidence of wound infections for particular surgeries [10,85]. Wound location near the groin in herniorrhaphy and prolonged (up to 5 days) catheter use can increase the risk of wound infection in CWI [84]. Catheters used for CWI can be clogged by blood or plasma with inflammation around the clot [86]. Catheter or pump failure happens in 1.1% [10], and is similar to epidural technical failure [87]. Wound breakdown and local complications are less likely in patients having WI [87]. Seroma can be expected during CWI, but was not reported in the meta-analysis [88].

## 6. Application of Wound Infiltration in Different Surgical Types

### 6.1. Cardiac Surgery

Pain after open cardiac surgery is primarily localized in the sternal and epigastric region, originating from the surgical incision and chest/mediastinal tubes, while prolonged intraoperative sternal retraction causes pain at the back of the shoulders [89]. As the pain is severe and aggravated by coughing, deep breathing and mobilization [90,91], effective analgesia is crucial to reduce pain, improve postoperative lung function, allow earlier extubation and decrease the incidence of pulmonary complications and cardiac dysrhythmias [89]. Various regional blocks have been tried as alternative to thoracic epidural analgesia in cardiothoracic surgery due to the risk of epidural hematoma after heparinization on cardiopulmonary bypass [92].

We identified 21 articles describing the use of WI or CWI after cardiac surgery. Usually, the surgeon performed WI [46,93,94,95,96,97,98,99,100,101]. LAs used to infiltrate the wound site included bupivacaine 0.5% [46,102,103], bupivacaine 0.25% [46,98,99], levobupivacaine 0.25% [95], levobupivacaine 0.25% with 1:400,000 epinephrine [96], ropivacaine 0.2% [93,94,104] and ropivacaine 0.5%, 0.3% [100]. Adjuncts to LA included off label use of magnesium sulfate [105] and tramadol [97]. Levobupivacaine seems to be the most suitable LA, with long duration, less cardiotoxicity potential than bupivacaine and wider availability than ropivacaine. Several studies examined the efficacy of infiltration versus saline placebo or no intervention in preventing pain after cardiac surgery [93,94,95,96,99,100,101,102,103,104]. WI applied alone [95] or combined with parasternal blocks [96] had significant opioid-sparing effect [95,96] and improved oxygenation at extubation [96]. Surgeons performed WI with levobupivacaine at the sternal incision and the mediastinal tube insertion site before sternal wire placement [95,96]. WI with levobupivacaine 0.25% resulted in effective analgesia with peak serum levobupivacaine concentrations below potentially toxic levels [96].

CWI is useful in cardiac surgery with insertion of one [97,99,101,102,103,106] or two catheters [93,94,98,100,104,107]. Duration of infusion ranged from 24 to 96 h [103,107], predominantly 48 h [46,93,94,97,98,99,101,102,104,106]. Prolonged bupivacaine infusion could be more effective in controlling delayed postoperative pain, but requires rigorous wound catheter care [103]. The bupivacaine infusion was started before sternal closure to provide an afferent block as early as possible.

Since the FDA advised against NSAIDs use after coronary artery bypass surgery and in patients with cardiovascular disease [108,109], opioids including morphine [94,95,97,100], oxycodone [100], piritramide [101] and fentanyl [105] are commonly used to supplement analgesia. Non-opioid analgesics used included paracetamol [93,94,97,105], metamizol [101], nefopam [97], ketoprofen [97], ketorolac [96,105] and tenoxicam [106]. Beside the opioid-sparing effect [46], CWI improved analgesia after cardiac surgery at rest [46,92,93,97,98,100,101,103,104,105] and during mobilization [93,94]. CWI enabled faster extubation [99,102,105], faster ambulation [46,94,107], improved satisfaction [46,93,94], reduced LOS [46,102,104,107], and reduced chronic pain 3 months after surgery [106], and seems beneficial as part of enhanced recovery after cardiac surgery protocols. In addition, one study demonstrated lower incidence of atrial fibrillation in the CWI group [102].

Although LAST has not been reported in these studies [93,94,99,100,101,104,107], further research is needed to determine the optimal type, concentration, and infusion rate of LA. In one study, ropivacaine concentration exceeded the safety threshold after 12 h of infusion, but there were no overdose symptoms [93]. Regarding location of the LA infusion catheters, placing sternal wound infusion catheters closer to the anterior branches of the intercostal nerves may improve analgesic efficacy. However, there is still a concern about catheter-related problems (e.g., accidental removal during dressing changes and breakage on removal) [46].

In total, 11 studies evaluated sternal wound infection during follow up, and showed no difference in incidence of wound infection or delayed healing in WI compared to control groups [93,99,101,104,105,106,107]. The incidence of sternal wound infection was 4.4–9.0% [97,100] and was lower than the group without a wound catheter [97]; however, the Agarwal et al. study showed higher incidence of sternal wound infections in CWI with ropivacaine compared to the historical group [100], and this finding led to premature discontinuation of the study [100]. Ropivacaine’s S-enantiomers and levobupivacaine have more significant immuno-supression potential than racemic bupivacaine [99,110]. One study showed that all wound catheter tips were sterile [101]. Handling of wound catheters should be similar to the handling of epidural catheters, including aseptic preparation of mixtures, rigorous hand hygiene, and aseptic, non-touch wound care [78].

Because published studies utilized diverse types of catheters (e.g., 5-inch soaker catheters, epidural catheters), anesthetic solutions, placement techniques (anterior to the sternum, subfascial and subcutaneous) and duration of CWI, expert agreement is needed for consistent use of WI techniques in cardiac surgery.

### 6.2. Thoracic Surgery

Thoracotomy is painful and involves multiple muscle layers, rib resection, and pain that intensifies with breathing movements [111]. In addition, acute post-thoracotomy pain intensity can influence the appearance and intensity of chronic post-thoracotomy pain [112]. Compared to open thoracic surgery, video-assisted thoracoscopic (VATS) procedures cause similar pain intensity in the first 24 h and similar incidence of chronic post-surgical pain [112]. The complexity of post-thoracotomy and post-thoracoscopic surgery pain necessitates perioperative multimodal analgesia, including use of regional analgesia in attempt to minimize opioid use [113].

Multiple studies investigated WI and CWI for analgesia after thoracotomy or VATS [31,114]. A retrospective study in open thoracotomy patients, compared thoracic epidural managed by the acute pain service vs. CWI placed by the surgeon combined with WI and intravenous opioid PCA [114]. Though maximum and average pain scores were higher in the CWI group, CWI was still a good option for post-thoracotomy analgesia, providing comfort, earlier discharge from the hospital and cost savings [114].

Before incision closure, WI with ropivacaine was safe in patients undergoing thoracotomy for esophageal cancer and, compared to placebo, resulted in better analgesia during 24 h, reduced postoperative analgesic (fentanyl, tramadol and flurbiprofen) consumption, earlier ambulation, higher patient satisfaction scores and shorter hospitalization [31]. However, a study comparing preoperative WI with 20 mL of 0.5% ropivacaine injected subcutaneously along the line of skin incision for thoracotomy and chest tube placement vs. preoperative ultrasound-guided erector spinae plane block (ESPB), showed superiority of EPSB, based on significantly reduced perioperative opioid consumption, better analgesia and reduced tramadol-related adverse events during 2 postoperative days [115].

With increasing popularity of VATS, it is encouraging that pre-emptive local WI with LA seems to be safe and effective as alternative to opioid intravenous (IV) PCA or other more invasive techniques for VATS major pulmonary resection [32,116]. Prospective, randomized trials studying pre-emptive WI in patients with palmar hyperhidrosis through bilateral needlescopic VATS for sympathectomy are interesting as each patient acted as their control [32,33]. Intramuscular diclofenac [32] and oral paracetamol and dextropropoxyphene [33] were used as supplemental analgesia. There was a trend for reduced pain on the side pre-treated with bupivacaine injection at the ports insertion site compared with contralateral side infiltration with placebo [33]. At 7 days after surgery, pre-emptive WI significantly reduced residual pain on the pre-treated side in 62.5% of the patients [33]. Additionally, another study investigating pre-emptive WI using lidocaine with epinephrine [32], reported significantly less pain 4 h and 24 h after surgery on the side treated with pre-emptive WI compared to the control side where epinephrine and normal saline were injected [32]. The clinical impact of this intervention is the possibility of early hospital discharge and early return to work with potential economic benefits [32]. However, paravertebral block provided better dynamic pain relief, reduced morphine consumption, and better patient satisfaction 24 h after VATS lobectomy than WI as part of multimodal analgesia with morphine and parecoxib [117].

At the present time PROSPECT does not recommend WI for thoracotomy and VATS due to lack of evidence, but clearly more research is needed [8]: WI or CWI as part of multimodal pain management after thoracotomy or VATS might present an option in fast-track surgery based on the logic that less invasive analgesia techniques should accompany less invasive surgery.

### 6.3. Abdominal Surgery

Epidural and bilateral peripheral nerve blocks have well-established benefits in abdominal surgery with midline incision. As laparoscopic approach to major abdominal surgery is becoming standard [6] there is opportunity for WI to emerge as alternative for various abdominal surgical procedures, especially the ones with midline incision [6,118,119]. Meta-analysis presented CWI efficacy comparable to epidural analgesia for different incision types like subcostal, midline or transverse incision [87]. Recovery parameters, opioid consumption, associated side effects and patient satisfaction seemed to be in favor of preperitoneal wound catheters compared to epidural analgesia for midline incisions and L-shaped incisions [26,30]. Local anesthetic adjuvants such as dexmedetomidine [120] or fentanyl [121] further increase efficacy without changing the incidence of postoperative nausea and vomiting. Cost-effectiveness analysis suggests that CWI is a promising strategy for managing postoperative pain compared to PCA-IV or epidural analgesia [122].

#### 6.3.1. Appendectomy

There are a limited number of studies on WI in patients undergoing appendectomy. The surgeon usually performs single-shot WI before incision [123,124,125] and rarely on wound closure. A study comparing WI before incision vs. after wound closure showed similar pain scores and opioid use during the first 48 h [126]. The most frequently used local anesthetic is bupivacaine 0.5% [124], bupivacaine 0.25% [127], lidocaine 1.5% with epinephrine [126] or their combination [123]. Lower pain scores and postoperative analgesic consumption were reported after WI with bupivacaine compared with no infiltration [124,125,127]. Postoperative systemic analgesia regimes included opioids like fentanyl [124], morphine [125] or meperidine [127] combined with nonsteroidal anti-inflammatory drugs ketorolac [124] or diclofenac [127]. The local anesthetic volume ranged from 10 to 15 mL in adults undergoing laparoscopic appendectomy [124,126]. Several studies comparing WI vs. placebo or no infiltration reported no difference in pain scores or postoperative opioid consumption after open appendectomy [126] with pre-incision WI [123] using lidocaine [126] or combination with bupivacaine [123]. WI did not influence wound complications [124].

#### 6.3.2. Laparoscopic Cholecystectomy

Laparoscopy is the chosen cholecystectomy approach as it is associated with less somatic pain; however, visceral pain originating from the gallbladder bed persists. Although there is low-level evidence that intraoperative local anesthetic WI can reduce acute pain after laparoscopic cholecystectomy, WI can be part of a multimodal pain management plan since adverse events are rare [128].

In laparoscopic cholecystectomy, timing of WI seems to matter: pre-incision single shot port site infiltration with ropivacaine (0.2%/0.5%/0.75%, 20 mL) provided analgesia comparable (regardless of concentration) to placebo [129], whereas single-shot trocar site infiltration with ropivacaine (1%, 20 mL) before skin closure lowered pain scores and analgesic use, but there was no difference in shoulder pain and nausea compared to placebo [130]. Bupivacaine peritoneal instillation before pneumoperitoneum creation added to pre-incisional trocar site infiltration produced adequate analgesia [131]. A randomized controlled trial comparing postoperative subcutaneous CWI with ropivacaine (0.75%) vs. saline showed that ropivacaine provided analgesia immediately and four hours after surgery but did not affect postoperative chronic pain [132].

Given the origin of visceral pain, one could expect that gallbladder bed infiltration with LA would provide analgesia. However, intraperitoneal administration of bupivacaine (0.5%,20 mL) was inferior to trocar site infiltration with the same amount of bupivacaine [133]. Addition of intraperitoneal instillation of lidocaine and bupivacaine to WI with 0.125% bupivacaine was not sufficient [134,135]. However, pre-incisional trocar site infiltration combined with infusion of high volume ropivacaine solution under the right hemidiaphragm at the beginning of surgery and saline infusion in the same location at the end of surgery, followed by rectal codeine and caffeine and oral ketoprofen reduced postoperative pain for 24 h compared to active and placebo control groups [136]. Gallbladder bed infiltration reduced visceral and shoulder pain, and trocar WI supplemented with intravenous ketorolac was superior for parietal pain for 24 h compared to no intervention [137]. However, WI combined with intraperitoneal ropivacaine administration at the end of surgery did not affect pain scores or time to hospital discharge in outpatient laparoscopic cholecystectomy compared to no intervention [138].

Apart from LAs, other medications used for WI include oxytocin or neosaxitoxine [139,140]. Interestingly, addition of clonidine (3 μg/kg) to bupivacaine had similar analgesic effect as when the same dose intravenous clonidine in addition to bupivacaine WI [48].

Compared to epidural analgesia, WI provided similar pain scores in the early postoperative period with lower cost after laparoscopic cholecystectomy [48]. However, thoracic epidural was associated with superior analgesia compared to single-shot WI with bupivacaine (0.5%,15 mL) administered before skin closure together with ketamine intravenous infusion in patients undergoing open cholecystectomy [141]. Compared to WI, bilateral ultrasound-guided transversus abdominis plane (TAP) block provided similar pain scores, lower opioid consumption, and higher patient satisfaction, but TAP was associated with three-fold increase in cost [142]. Patients with laparoscopically delivered TAP had lower pain scores at rest and cough during the first 6 postoperative hours, but no difference in shoulder pain compared to patients receiving periportal bupivacaine infiltration [143].

#### 6.3.3. Inguinal Herniorrhaphy

Acute postoperative pain after inguinal herniorrhaphy is a complex symptom encompassing both somatic and visceral component. PROSPECT recommends WI alone or in combination with sedation or general anesthesia for inguinal herniorrhaphy [8]. Patients receiving pre-incisional single shot WI bupivacaine (0.25%) had similar pain scores, analgesic consumption, and overall patient satisfaction as patients receiving placebo infiltration with saline [144]. Compared to placebo, single-shot WI with bupivacaine (20 mL, 0.5%, 0.25%) at the end of surgery with diclofenac [145] and tramadol [146] as additional analgesia provided lower pain scores at rest and on movement and lower analgesic consumption during the first 4 hours [146] to 24 h after surgery [145].

Pre-incisional single-shot WI using different lidocaine concentrations (0.25%, 0.33% and 0.5%) or bupivacaine (0.25%) were not significantly different with regards to intraoperative pain scores, patient satisfaction, analgesic consumption or incidence of adverse events compared to placebo [144,147]. Levobupivacaine and racemic bupivacaine as single shot WI showed similar analgesic efficacy [148].

Variations in delivery model and type of medication influence the effect of wound infiltration: compared to placebo, CWI with bupivacaine (0.5%) for 48 h after open inguinal herniorrhaphy reduced opioid use and pain with no apparent increase in wound-related complications [149]. Implementation of bupivacaine infused collagen-matrix implant resulted in improved postoperative analgesia and lower opioid use for up to 72 h compared to placebo [150]. Single-shot WI before skin closure with tramadol (1 mg/kg) reduced pain scores and analgesic use compared to WI with bupivacaine, but this difference could be attributed to systemic resorption of tramadol [49,151]. WI with meloxicam (7.5 mg) offered no efficacy advantage over systemic administration, but could potentially elicit fewer systemic adverse events [50].

#### 6.3.4. Esophagogastric Surgery

Esophagogastric surgery is a part of treatment for malignancies or morbid obesity. CWI with ropivacaine (0.3%,5 mL/h) after open gastrectomy provided comparable efficacy to continuous epidural analgesia and opioid-based PCA-IV, lowered morphine consumption, reduced postoperative nausea and vomiting, and enabled earlier bowel recovery and shorter LOS [152].

Special patient populations may significantly benefit from WI after esophagogastric surgery. Geriatric patients undergoing laparoscopic gastrectomy who received single shot WI with bupivacaine (0.5%, 40 mL) had lower postoperative pain scores and lower morphine consumption for 48 h compared to placebo [153]. In bariatric patients, WI could be a prudent opioid-sparing option [154]. However, single-shot pre-incision WI bupivacaine (0.5%) with epinephrine was not an effective analgesic strategy for patients undergoing laparoscopic bariatric surgery [155]. Dexmedetomidine as adjuvant to ropivacaine enhanced the analgesic efficacy of ropivacaine WI, reduced 24-hour sufentanil consumption and had no adverse effect on wound healing in patients undergoing open gastrectomy [156].

Ultrasound-guided TAP with rectus sheath block provided superior analgesia compared to WI in patients undergoing major upper abdominal surgery [157]. Currently available data suggest that WI is not associated with increased incidence of wound complications [156,157].

#### 6.3.5. Hepatic, Biliary, and Pancreatic Surgery

Compared to placebo, both continuous and single-shot ropivacaine WI resulted in lower pain scores, reduced opioid consumption, reduced stress hormones levels, shorter LOS, and faster bowel recovery after open hepatectomy [158,159,160]. CWI showed equivalent efficacy as epidural PCA and opioid intravenous analgesia after open hepatectomy [161,162]. In patients undergoing laparoscopic hepatectomy, WI and ropivacaine infused gelatin sponge placed on the liver cutting surface provided lower pain scores at rest and on movement, reduced opioid consumption, and lower stress hormones levels during 48 h compared with placebo [163]. Meta-analyses showed comparable pain scores on the second and third postoperative day between CWI and epidural analgesia, except significantly higher pain scores on a postoperative day one after open liver resection with conflicting conclusions regarding functional recovery [164,165]. In open hepatic resection, CWI has significant potential advantage compared to epidural analgesia, in terms of lower incidence of perioperative hypotension, lower vasopressor use and better safety profile in cases of postoperative coagulopathy during 48 h follow up [166]. WI was not associated with wound-related complications in patients undergoing liver resection [163,166]. In conclusion, single-shot or CWI with local anesthetic as part of multimodal pain therapy can be useful alternatives to epidural analgesia in patients undergoing open or laparoscopic hepatic surgery.

#### 6.3.6. Colorectal Surgery

Colorectal surgery has seen a major shift from open to laparoscopic techniques in recent years. Compared to open surgery, laparoscopic colorectal surgery results in similar visceral acute postoperative pain, whereas the parietal component of postoperative pain is significantly different, resulting in overall lower pain intensity on mobilization [167]. Compared to placebo or routine analgesia, WI appears to reduce opioid requirements and pain scores and improves recovery after colorectal surgery [87,168]. CWI with ropivacaine supplemented with postoperative ketoprofen and paracetamol, reduced morphine consumption for 72 h, improved pain relief at rest for 12 h and with cough for 48 h, and accelerated postoperative recovery compared to placebo in open colorectal surgery [23]. Additionally, liposomal bupivacaine is associated with lower cost of overall postoperative pain management compared to control after laparoscopic colorectal surgery [169] and reduced pain and opioid requirement through 72 h after hemorrhoidectomy [170].

In patients undergoing laparoscopic colon resection, CWI ropivacaine combined with systemic ketorolac and propacetamol after surgery showed similar efficacy, postoperative inflammatory response, incidence of wound-related complications, and cancer recurrence in comparison to PCA-IV opioid during 48 h [171]. No difference in CWI efficacy was observed between ropivacaine and lidocaine for 48 h [172]. Single-shot WI with bupivacaine at the end of laparoscopic single-incision colectomy resulted in lower pain scores and lower analgesic consumption compared to no intervention [173].

Pain relief with CWI was equal to thoracic epidural analgesia for 72 h after open colorectal surgery [174]. Single shot WI could be successfully supplemented by TAP block, ketorolac and paracetamol to reduce pain score, nausea, and vomiting and accelerate bowel function after laparoscopic colorectal surgery [175]. The skill of the TAP block provider was crucial for regional block success in studies comparing TAP block vs. CWI [176]. Single-shot WI provides comparable short-term postoperative analgesia as TAP block, but TAP block has better long-lasting effect [177,178]. PROSPECT recommends CWI as epidural substitute for open colorectal surgery [8]. In colorectal surgery, WI did not impact wound-related complications [23,173,179], and did not influence chronic postoperative pain for up to one year after surgery [171]. Evidently, the role of CWI in laparoscopic colorectal surgery deserves further investigation.

#### 6.3.7. Reconstruction of the Abdominal Aorta

Use of single-shot WI or CWI in reconstructive abdominal surgery has not been adequately explored. However, WI analgesia can be helpful in emergency cases of ruptured abdominal aneurysm where there is no time for epidural catheter placement. An open label, non-inferiority randomized trial in patient undergoing open abdominal aortic aneurysm repair showed that CWI with levobupivacaine combined with PCA-IV morphine and paracetamol provided analgesia comparable to continuous epidural analgesia, but patients in the CWI group had inferior early pain control and required higher doses of rescue IV morphine during the first 4 to 48 h after surgery [180].

### 6.4. Breast Surgery

Breast surgery is an umbrella term used to describe various procedures ranging from simple biopsies performed in minutes with minimal scarring to radical mastectomy with lymph node dissection, which is a traumatic, mutilating operation. Since breast operations usually are outpatient procedures or require short hospital stay, most WI analgesia studies focus on acute postoperative pain. Current guidelines suggest WI and paravertebral or pectoral muscle blocks for major oncological breast surgery [9]. However, there are limited and conflicting data from high-quality randomized, controlled studies suggesting that WI is a reliably effective analgesic [181].

Single-shot local anesthetic WI during breast cancer surgery showed modest reduction of pain in the first few hours after surgery but did not reduce postoperative analgesic consumption [81,88]. CWI combined with systemic paracetamol, nefopam and ketoprofen was associated with reduced pain intensity and morphine consumption during postoperative 24 h compared with placebo [36,182].

Most studies, however, evaluated the efficacy of a single shot local anesthetic WI compared to placebo or general anesthesia alone. Single-shot WI was performed by the surgeon, usually at the end of surgery. Pre-incisional WI is reported scarcely and with disappointing results [183,184]. Intraoperative WI ropivacaine (0.375% or 0.75%) provided lower VAS scores at rest and on mobilization 90 min to 6 hours after surgery compared to placebo [185,186]. Compared to no infiltration, single-shot WI with bupivacaine (0.25%, 10 mL) provided better pain relief, lower analgesic consumption for up to 16 h [187] and lower opioid consumption for up to 48 h after surgery [83]. As part of multimodal analgesia, pre-incision WI with lidocaine (1%, 10 mL) and bupivacaine (0.5%, 10 mL) combination, followed by post-resection injection of 7 mL in the breast incision site plus additional 3 mL in the sentinel node incision site provided opioid-free analgesia after oncological breast surgery compared to patients without multimodal analgesia [184].

Few studies compared WI to other regional techniques, including paravertebral block and serratus plane block. CWI with ropivacaine provided better analgesia even during movements than a single-shot paravertebral block, but had higher incidence of postoperative nausea and vomiting during 24-hour follow up [188]. Single-shot WI with bupivacaine (0.25%, 10 mL) provided similar pain scores compared to continuous paravertebral block up to 48 h after surgery [189]. However, WI with bupivacaine or levobupivacaine with epinephrine was inferior to ultrasound guided paravertebral block or serratus plane block in the first 24 h after surgery [190,191]. Due to significant variability in reported regional techniques, further research is needed to adequately compare the efficacy and safety of these techniques.

A completely different approach was taken in esthetic surgery: Two observation studies without a control group showed that tumescent local anesthesia for the breast surgery was associated with moderate pain relief [192,193]. Ultrasound needle guidance [192] during LA injection assured the efficacy of WI anesthesia before incision and repetition during surgery [193].

The LA most frequently used is ropivacaine [182,185,186,194] followed by bupivacaine [83,187,189,190], levobupivacaine with epinephrine and clonidine [191] and lidocaine [192,193] and mixture of lidocaine and bupivacaine [184]. Adding fentanyl to ropivacaine did not provide any benefit [195]. Single WI or CWI did not reduce the incidence of chronic postoperative pain after 6 and 12 months [182,186,194].

A variety of LAs, volumes, concentrations, and techniques are used for WI in breast surgery. Most frequently, studies compare WI efficacy to general anesthesia alone or placebo infiltration but seldom to other regional techniques. Available data suggest reduced pain scores and analgesic consumption as benefits associated with WI up to 24 h after surgery. WI for breast surgery is not associated with increased prevalence of postoperative complications, except for superficial bruising [83].

### 6.5. Thyroid Surgery

Authors rarely explore the WI’s effectiveness in thyroid surgery, and results are seldom comparable due to heterogeneity in study design and medication selection. WI with bupivacaine (0.5%, 10 mL) reduced postoperative pain scores and analgesic consumption up to 24 h after surgery compared to no infiltration at all [196,197] or placebo [198]. Single-shot WI with ropivacaine (0.75%) at the end of thyroid surgery did not show any significant analgesic benefit compared to placebo [199]. However, thyroid surgery can be performed with lidocaine infiltration of the incisional site and sedation [200].

The addition of NSAIDs like lornoxicam (8 mg) to ropivacaine (0.75%) improved postoperative pain control and patient comfort and decreased the need for postoperative opioids during 4 postoperative hours compared with ropivacaine and lornoxicam alone, and 12 h compared to placebo [51]. WI with diclofenac (50 mg) reduced pain scores and rescue analgesic (tramadol) use during the first 24 h postoperatively compared to bupivacaine (0.25%, 10 mL) [52]. Although superficial cervical plexus block is the most frequently used regional technique, bilaterally performed WI has similar efficacy with lower incidence of transient mild adverse events during 24 h [201,202]. Single pre-incision WI with bupivacaine did not affect wound healing compared to no infiltration [197]. We could not find any data evaluating CWI during or after thyroid surgery.

### 6.6. Neurosurgery

Neurosurgical procedures, especially craniotomy, can result in pain that ranges from moderate to excruciating [203] in 40–84% of patients in the first 12 h after surgery [204]. Possible causes of suboptimal postoperative pain relief in neurosurgery patients include the need for prompt neurologic assessment after brain surgery, lack of robust evidence comparing different analgesics, and patient inability to express pain verbally [203]. Undertreated pain after craniotomy may cause adverse consequences, including hypertension and postoperative intracerebral hemorrhage [204].

Pain after craniotomy originates from pericranial muscle and soft tissue. Suboccipital and subtemporal interventions are associated with high incidence of pain [205]. Non-sedating analgesic options, including scalp blocks and WI, are technically more comfortable and tolerable for the patient when performed before incision or at the end of the operation. The standard route local anesthetic administration in patients undergoing brain surgery is scalp infiltration, is not related to any specific sensory pathways. Scalp block was superior to WI of the pin insertion sites based on lower postoperative pain scores, longer time to first analgesia request, lower incidence of postoperative nausea and vomiting [206], and lower plasma cortisol and adrenocorticotropic hormone 5 and 60 min after surgery [207].

Most published studies on WI in neurosurgery included patients undergoing supratentorial craniotomy [208,209,210,211], while one study included patients undergoing infratentorial surgery [205]. WI can be done by surgeons [210,211,212], anesthesiologists [207,213], or both [206]. LA used to infiltrate around the surgical wound site included bupivacaine 0.5% [205,207] or 0.25% [210,214], bupivacaine 0.375% with 1:200 000 epinephrine [215], bupivacaine 0.5% with epinephrine [208], ropivacaine 0.75% [206,215], and 0.5% [211], 0.5% ropivacaine and 1% lidocaine [212], 0.5% bupivacaine and 2% lidocaine with 1:200 000 epinephrine [213]. Most studies compared the efficacy of WI vs. saline placebo [205,208,210,211,214,215] or no intervention in preventing pain after craniotomy [209].

WI’s efficacy for treating acute pain after neurosurgery is controversial, probably because of study heterogeneity. Scalp infiltration was performed mostly before surgical incision [205,206,207,208,210,211,213,214]. Scalp infiltration has been reported as effective analgesia method if used pre-incision [203], before pinning [205], before skin closure [208] and at the end of surgery [209]. Additionally, scalp WI performed before surgical incision showed better results compared to infiltration performed at the end of surgery before skin closure [203]. Duration of postoperative analgesia ranged from 1 to 6 hours, and in one study up to 24 h [203,208,209].

Several studies have measured the quantity of additional analgesia consumption [205,209,210,211,215]. Opioids used as main analgesics [216] after craniotomy include morphine [203,205,211,215], fentanyl [210], tramadol [203], nalbuphine [209] and oxycodone [206]. Additional non-opioid analgesics included paracetamol [203,209], tenoxicam [214] and diclofenac [210]. Use of NSAIDs, including COX-2 inhibitors in neurosurgery demands further investigation regarding benefits and safety [216]. Pre-incisional WI showed opioid-sparing effects [203,211], but there was no difference in LOS in one study [211], and we could not find data on ICU LOS. One study presented a lower number of patients with persistent postoperative pain 2 months after surgery [209].

Nausea and vomiting have been reported by seven studies [203,205,208,209,211,214,215]. Less common adverse events included hypotension, hypertension, bleeding, delirium, visual disturbances, agitation, respiratory depression, pruritis, diarrhea, and constipation.

### 6.7. Urology

Although open nephrectomy is associated with severe postoperative pain, WI is rarely explored in the literature [217,218]. Compared to epidural analgesia, CWI as component of multimodal analgesia showed slightly higher pain scores on the first and third postoperative day and higher need for supplementary analgesia (tramadol) after open renal surgery [217]. In this study, as in others recently published, single WI preceded CWI [152,180,217]. CWI potentially presents safety advantages compared to epidural analgesia because of lower risk of neurological complications [219].

A retrospective study on 1458 patients compared WI vs. intercostal nerve block at the end of surgery using combination of bupivacaine and lidocaine, and tramadol as supplementary analgesia after flank incision for open nephrectomy and other procedures involving renal pathology [218]. Although both techniques were effective, WI provided better pain control with lower total tramadol use and lower cost for 72 h after surgery [218]. However, single-shot WI in more extensive surgeries has inconsistent results [220].

Interestingly, the PROSPECT group recommends WI at the end of surgery in open prostatectomy and at the port insertion site in video-assisted prostatectomy [8], and based this recommendation on “transferable data” from herniorrhaphy and laparoscopic cholecystectomy, because of technical suitability and good WI safety profile [8].

### 6.8. Gynecological Surgery

We identified 18 studies investigating WI in different gynecological procedures. Most studies were placebo controlled [221,222,223] and one compared liposomal bupivacaine with 0.25% bupivacaine [224]. PROSPECT recommends WI for elective Cesarean section and abdominal hysterectomy [8]. CWI with ropivacaine provided similar analgesic effects as PCA fentanyl and ketorolac after laparoscopic gynecologic surgery, and despite higher rescue analgesic use, benefits included opioid-sparing effects and fewer side effects during 24 h follow up [225]. Single WI with levobupivacaine [221,222], bupivacaine [223], or liposomal bupivacaine [224] in addition to general anesthesia and standard analgesic therapy including NSAIDs or paracetamol and opioids significantly decreased postoperative analgesic requirement [221,222,223,224], lowered pain intensity [221,222,224] and reduced time to ambulation after laparoscopic [221,222,224] and open gynecological surgery [223]. The effects lasted for several to twelve hours [221].

Compared to TAP, single WI showed inferior analgesia [226,227,228]. However, CWI as part of multimodal management showed better [229] or similar analgesic effect as PCA-IV fentanyl [225], and this finding might be important in cancer surgery patients [229]. One meta-analysis showed, that compared with bupivacaine alone, addition of ketamine or dexmedetomidine to bupivacaine for WI showed opioid-sparing effect, delayed first request for rescue analgesia, and attenuated postoperative stress response in total abdominal hysterectomy [54]. Pre-incision port site infiltration with liposomal bupivacaine compared with bupivacaine decreased pain on the second and third postoperative day after laparoscopic or robotic multiport hysterectomy [224]. Surgical approach may influence postoperative pain when WI is used, as patients needed less opioid after laparoscopic gynecological surgery compared to transabdominal surgeries [230]. WI seems to be a valuable addition to analgesia, especially after gynecological oncological surgeries. Quality randomized controlled trials are needed in search of the best type of local anesthetic, adjunct, and technical approach in gynecological surgery.

### 6.9. Orthopedic Surgery

WI is a frequent addendum to other regional techniques for different types of orthopedic surgical procedures, and it is widely presented in the literature [6,231,232,233]. WI’s popularity in orthopedics can be explained by the flexibility of the technique, ability to provide early mobilization, and safety, which is particularly desirable in geriatric patients and patients with multiple comorbidities [232]. PROSPECT recommends WI with local anesthetics for laminectomy before wound closure and as alternative to ankle block for hallux valgus surgery [8].

Novel studies suggest improved WI efficacy by adding NSAIDs or epinephrine or combining single WI and CWI [53], resulting in improved analgesia during early mobilization. Although these are off-label uses of NSAIDs, side effects were not reported in any of these studies; WI with ketorolac, levobupivacaine and epinephrine enabled better mobilization, shorter duration of physical therapy, reduced PCA-IV opioid use, and reduced LOS compared to WI with local anesthetic chosen by surgeon after spine surgery [53].

In total hip replacement, combination of spinal anesthesia, CWI with levobupivacaine and local infiltration analgesia next to the implant, fascial and subcutaneous tissues was compared with placebo [234]. The follow up period was 72 h and additional analgesics included ketorolac and morphine [234]. This multimodal approach resulted in better analgesia, decreased number of analgesia requests and improved physical therapy with less pain [234]. Although no infection was detected in this study, the authors emphasized strict use of aseptic techniques during catheter placement and care because of proximity to artificial implant material [234]. However, the analgesic benefit of WI has been questioned by a small recent RCT that showed no analgesic benefit with injecting ropivacaine vs. normal saline [235].

In open reduction and internal fixation (ORIF) of ankle fractures local infiltrative analgesia accompanied with PCA-IV morphine provided better pain scores at the eighth hour, opioid-sparing effect, and fewer side effects during 48 h follow up compared to PCA-IV alone [236]. As liposomal bupivacaine (LB) offers analgesia for up to 72 h, avoidance of continuous infusion catheters makes it desirable for postoperative analgesia in orthopedics [237]. A panel of expert anesthesiologists and surgeons recommended using 120 mL (20 mL of LB, 20 mL bupivacaine 0.25% and 80 mL saline) for extracapsular procedures and 80 mL (20 mL of LB, 20 mL bupivacaine 0.25% and 40 mL saline) for intracapsular procedures, using 22-gauge needle and small volume injections using tracking or combination with fanning technique in hip surgery [238]. In a retrospective study on patients undergoing hemiarthroplasty for femoral neck fractures, patients who received periarticular LB injection as part of multimodal pain management had comparable pain control but reduced need for ICU care, significantly shorter LOS and higher probability to be ambulatory at discharge compared to no infiltration [239].

Addition of local infiltration analgesia with ropivacaine after knee surgery resulted in adequate analgesia, better mobilization on the first day compared to nerve blocks and good muscle strength for up to 3 days [240]. Intraoperative periarticular local infiltration analgesia compared with placebo or no infiltration might be helpful as analgesia for the first 24 h after total knee arthroplasty [241]. Two meta-analyses show that compared to epidural analgesia, local infiltration analgesia increases range of motion, shortens LOS, and lowers nausea and vomiting incidence after total knee surgery [241,242]. Periarticular injection of bupivacaine combined with ketorolac and epinephrine, given once during total knee arthroplasty and twice intermittently in the postoperative period showed lower pain scores, earlier mobilization and reduced LOS compared to subarachnoid morphine [243]. Use of liposomal structures not only for bupivacaine, but also for NSAIDs, decreases inflammation after local injection, improves NSAIDs’ effectiveness and minimizes side effects [244]. WI with LB as part of multimodal pain therapy resulted in equal analgesia with opioid-sparing effect compared with continuous femoral nerve block in patients undergoing total knee arthroplasty [245]. One meta-analysis showed modest difference between local infiltration analgesia and peripheral nerve blocks in analgesia quality and opioid consumption 24 h after total hip arthroplasty, and the authors suggested that the cost and side effects of these techniques need further analysis [246]. Periarticular injection of LAs (bupivacaine) provided analgesia quality similar to peripheral nerve blocks for shoulder surgery with significant opioid-sparing effect and reduced side effects [247]. Liposomal bupivacaine is also used for foot and ankle surgery [232]. Local infiltration analgesia, WI and CWI are viable alternatives when peripheral nerve blocks cannot be performed due to lack of staff or equipment [248], when motor block is undesirable and there is need for immediate mobilization [5,240], and in patients with coagulation abnormalities or on anticoagulation therapy (with the exemption of compressible sites where peripheral nerve blocks are not contraindicated) [3,249].

### 6.10. Ambulatory Surgical Procedures

Beside the above mentioned applications of WI for breast surgery, herniorrhaphy, and orthopedic surgery, WI is widely used in ambulatory plastic surgery and varicose vein surgery. However, single-dose bupivacaine WI provided analgesia after bilateral saphenofemoral junction ligation for varicose veins only in the immediate postoperative recovery phase [82].

### 6.11. Trauma and Emergency Surgery

Three-quarters of major trauma victims will experience moderate-to-severe pain due to their injuries or the management of these injuries [250,251]. Poorly treated pain can result in considerable psychological stress, impacting ongoing treatment and post-injury rehabilitation. Adequate analgesia reduces the adverse effects associated with undertreated pain [250]. The efficacy of multimodal pain interventions in nonelective trauma procedures has been assessed in specific subgroups like orthopedic surgeries [252], but remains incompletely evaluated in other types of surgery. WI may be beneficial after abdominal exploration and can be a useful adjunct for postoperative pain control in the trauma patient, thereby limiting the adverse effects of systemic opioids.

## 7. Wound Infiltration in Enhanced Recovery after Surgery Protocols

The enhanced recovery after surgery (ERAS) is the gold standard in contemporary surgical practice aiming to reduce stress, speed patient recovery, and return to daily activities. The use of multimodal analgesia is a postulate of ERAS protocols with elimination and reduction of opioids use and consequent promotion of early mobilization, bowel motility, the prevention of nausea and vomiting, and long-term consequences of opioids use [253]. Thus, regional analgesic techniques that include neuraxial (e.g., epidural, spinal), peripheral nerve blocks, and wound infiltration are part of current ERAS protocols.

Recent guidelines for enhanced recovery after lung surgery suggest multimodal analgesia, including regional analgesia or local anesthetic techniques, in an attempt to avoid or minimize opioids and their side effects [113]. ERAS protocol updates need to promote the use of WI in VATS, where current evidence suggests that WI is very effective [113]. Guidelines for ERAS after cardiac surgery do not include WI [254], but further research is needed in this field. Similarly, esophageal surgery ERAS protocols do not mention WI as an analgesic option [255], whereas the ERAS Society recommends WI with LA particularly with ropivacaine or levobupivacaine [256] after bariatric surgery (high evidence level, strong grade of recommendation). In addition, pre-incision WI [136] combined with intraoperative bupivacaine aerosolization [257] may present a reasonable option for enhancing recovery after bariatric surgery [256]. Although there are no clear recommendations about safe doses of LAs in bariatric surgery ERAS protocols, doses of local anesthetic should be calculated based on patient’s ideal body weight (IBW), in order to reduce the risk of LA toxicity.

Although published studies support the use of CWI or WI in open colorectal surgery, current ERAS protocols do not recommend its use [258]. ERAS recommendation for rectal/pelvic surgery states that there is low evidence level and therefore weak recommendation for CWI via pre-peritoneal catheters due to “limited evidence” from ERAS protocol-based studies [259]. However, there is clear recommendation for CWI through preperitoneal catheter as “alterantive to epidural” in ERAS for open pancreaticoduodenectomy (high evidence level, strong grade of recommendation) [260].

ERAS protocol for major head and neck cancer surgery with free flap reconstruction recommends only systemic analgesia [261]. In neurosurgery, although scalp infiltration and scalp blocks can be recommended for craniotomies, there is no ERAS Society protocol due to lack of evidence [262,263].

ERAS protocols in urology recommend epidural analgesia for open abdominal and pelvic procedures [264]. However, available data suggest the use of CWI with pre-peritoneal catheters combined with systemic analgesia (paracetamol and NSAIDs) for minimally invasive surgical procedures instead of different types of regional analgesia and intravenous lidocaine [264]. The recent update of ERAS for gynecological procedures recommends WI with bupivacaine (high evidence level) while noting that studies are needed to compare thoracic epidural analgesia vs. transversus abdominis block and WI [265].

In orthopedic surgery, ERAS recommends LIA with LA for knee replacement (evidence level high, recommendation grade strong), but not for hip replacement [266]. The authors explain that the advantages of LIA over peripheral nerve blocks and neuraxial blocks include the absence of motor blockade, thus enabling early mobilization, the preservation of hemodynamic stability and the absence of influence on urine retention [266]. In vascular surgery, a recent systematic review suggested that use of ERAS protocols is currently limited because of low quality evidence regarding feasibility and effectiveness [267]. One RCT comparing thoracic epidural analgesia vs. CWI analgesia after abdominal aortic surgery showed that CWI resulted in comparable, good pain control but required higher doses of LA [268]. Because there is potential for WI or CWI to be beneficial, additional high-quality studies are needed to evaluate WI and CWI as part of multimodal recovery after vascular surgery [267]. We believe that WI can be an important addition to ERAS protocols, and has the potential to be a low cost, easy and safe alternative to other techniques currently in ERAS protocols, such as nerve blocks or neuraxial analgesia.

## 8. Future Directions

Literature data analysis and our experience helped us identify possible practice improvements. More research is needed on the role of liposomal bupivacaine in WI, particularly exploration of its efficacy and cost. Also, WI should be considered in emergency surgeries like abdominal aortic repair when regional blocks are impractical before incision.

We also propose the combination of pre-incisional single-shot WI and postoperative CWI especially in abdominal surgery. Based on unpublished experience from the European Pain Federation (EFIC) supported international quality improvement project under the name PAIN OUT, we propose implementation of WI as part of multimodal management whenever possible, particularly when other regional anesthesia techniques are contraindicated. The limited duration of analgesia provided with WI necessitates the implementation of other modalities of care before the LA’s effect is over.

ERAS protocols strongly recommend intravenous lidocaine infusion for open and laparoscopic colorectal surgery [258], due to opioid-sparing effect, with reduced incidence of nausea, vomiting [269] and postoperative ileus [270]. Published data suggest that toxicity of intravenous lidocaine is related to plasma concentration, and although toxicity is infrequent, monitoring patients for signs of toxicity is mandatory [271]. Since lidocaine infusion has been used in genitourinary, gynecology, ambulatory, breast, spine and cardiothoracic surgery [270], the idea of combining IV lidocaine and WI with LA sounds tempting. However, because the pharmacokinetic profile of combining IV lidocaine infusion with LA WI has not been studied, safety is a major concern as it is reasonable to expect a higher incidence of toxic effects.

We propose the use of a check list for single shot WI, which can be adjusted for CWI (Figure 3). As our literature search revealed that most studies use “one size fits all” LA dosage regimens, individually calculated LA doses need to be evaluated in future studies.

A personalized WI plan improve postoperative analgesia, by individualizing LA concentration and infusion rate, as well as catheter position depending on incision. Protocols proposed by expert groups will need to include WI as part of multimodal analgesia. Last, additional research is needed on potential WI immuno-modulatory effects, especially in oncologic surgery.

## 9. Conclusions

WI and CWI are simple, practical steps in a multimodal approach to postoperative analgesia. WI requires less time and equipment and is cheaper, faster, and more acceptable to surgeons. CWI is superior to WI in terms of prolonged action, significantly reduced postoperative opioid consumption and opioid side effects, thus accelerating postoperative recovery. As access to educational material is becoming easy, patients are better informed about treatment options, and surgeons are increasingly aware of the importance of quick painless recovery after surgery. The economic benefit of fast recovery and return to work are also important. Furthermore, WI may present a valuable analgesia component in fragile patients, such as geriatric patients, obese patients and patients with multiple comorbid states or chronic pain.

WI techniques have a low incidence of complications. They are simple and quick to perform, easy to manage, have opioid-sparing effect, and have no major contraindications, other than patient refusal or local infection. As new studies document the safety of infiltrative techniques, surgeons will likely accept and promote more frequent WI use based on the type of surgical procedures and individual patient needs.

## Figures and Tables

**Figure 1 jcm-10-04659-f001:**
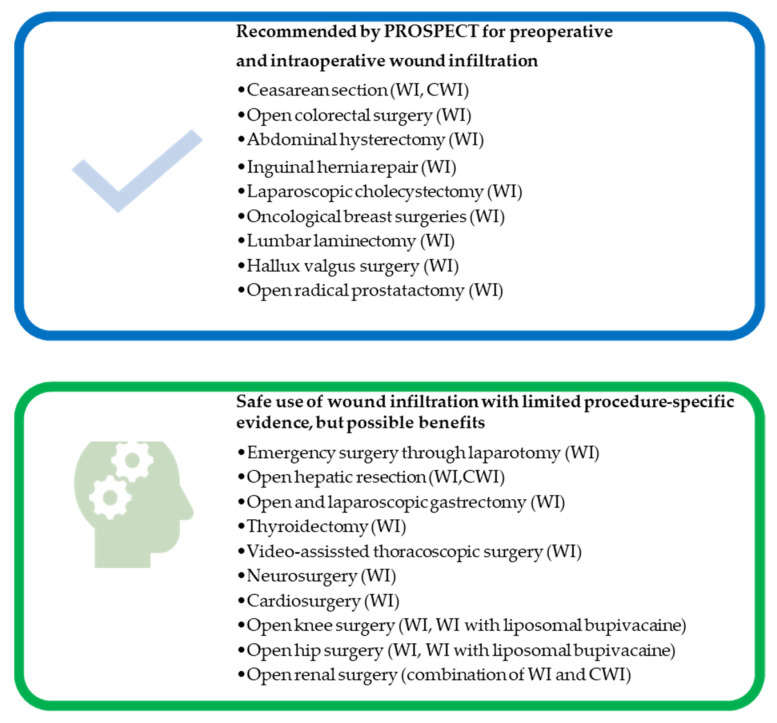
Summary of wound infiltration (WI) application in the preoperative and intraoperative period. European Society of Regional Anesthesia procedure-specific postoperative pain management (PROSPECT) working group recommended WI as part of multimodal pain management for several types of surgical procedures. Published studies suggest that WI is appropriate and safe when other techniques are contraindicated, but requires further investigation for efficiency in different kind of surgeries. WI-wound infiltration; CWI-continuous wound infiltration.

**Figure 2 jcm-10-04659-f002:**
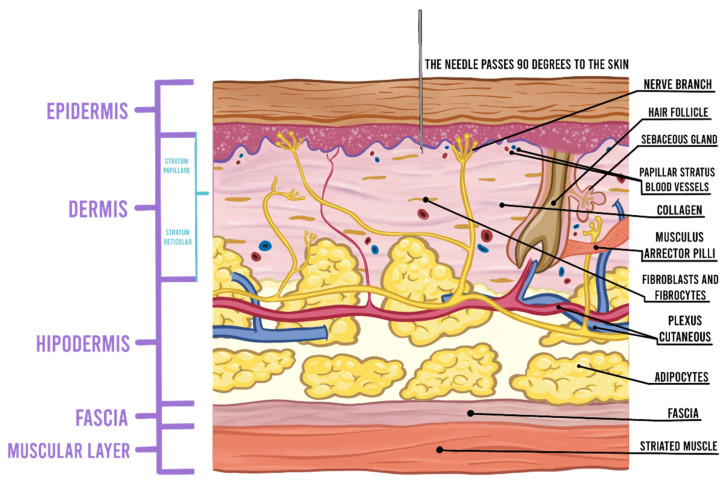
Needle insertion at a 90-degree angle during skin penetration passes through fewer pain fibers and reduces injection pain.

**Figure 3 jcm-10-04659-f003:**
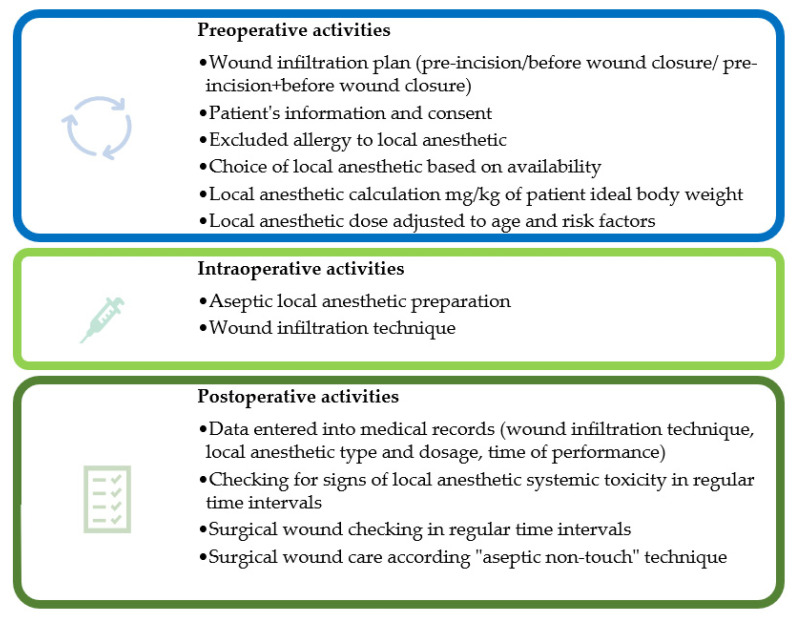
Check-list for single-shot wound infiltration planning, performance, and follow-up.

**Table 1 jcm-10-04659-t001:** Recommended local anesthetics doses for adults.

Local Anesthetic	Adult Dosing without Epinephrine	Adult Dosing with Epinephrine	Duration without Epinephrine (min)	Duration with Epinephrine (min)	Strength of Recommendation	Level of Evidence
Lidocaine [59]	4.5 mg/kg (max:300 mg)	<7 mg/kg (max 500 mg)	30–120	60–400	C	III
Mepivacaine [59,60]	6 mg/kg (max < 300 mg)	7 mg/kg (max < 500 mg)	30–120	60–400	No data	No data
Bupivacaine [59,60,61]	2 mg/kg (max 400 mg)	3 mg/kg (max 225 mg)	120–240	240–480	No data	No data
Ropivacaine [59]	2.9 mg/kg (max 200 mg)	-	No data	No data	No data	No data
Procaine [59,60]	10 mg/kg (max 350–500 mg)	16 mg/kg	15–30	30–90	No data	No data

**Table 2 jcm-10-04659-t002:** Summary of recommended local anesthetics doses by type of surgery where continuous wound infiltration is used.

Surgery	Catheter Location and Type	Pre-Closure Infiltration	Local Anesthetic	Postoperatively Dosing Regimen	Duration of Infusion *
Shoulder surgery [4] Arthroscopic Open	Subacromial (High) Multi-orifice/epidural (Moderate)	-	Bupivacaine 0.25% Ropivacaine 0.2% (Moderate)	2–5 mL/h (High)	48 h (Moderate)
	Subacromial Multi-orifice (Low)	Ropivacaine 0.5%/0.75%, 30 mL [65] (Low)	Ropivacaine 0.375% (Low)	5 mL/h (Low)	48 h (Low)
Knee surgery [4] Anterior cruciate ligament reconstruction Total knee arthroplasty	Intra-articular/combination with subcutaneous (Low) Multi-orifice (Moderate)	-	Bupivacaine 0.25% Ropivacaine 0.25% (Moderate)	4–10 mL/h (Moderate)	48 h (Moderate)
	Intra-articular/combination with subcutaneous parapatellar area (Low) Multi-orifice (Low)	Ropivacaine (0.2%) + epinephrine (1 mg/mL) + ketorolac (30 mg/mL) WI along all layers [66] (Low)	Ropivacaine 0.2% (Low)	5 mL/h (Low)	48 h (Low)
Hip surgery [4] Total hip arthroplasty Minimally invasive approach to total hip arthroplasty	Subcutaneous all along wound + epicapsullary (Double catheter technique) (Low) Multi-orifice (Low)	Ropivacaine 0.3%, 20 mL (Low)	Ropivacaine 0.2% (Low)	5 mL/h (Low)	48 h (Low)
	Epicapsullary (Low) Multi-orifice	-	Ropivacaine 0.3% (Low)	8 mL/h (Low)	48 h (Low)
Spine surgery [4] Iliac crest bone harvesting	Above the fascia Double catheter technique- “one catheter tip opposite to other” (Low) Multi-orifice (Low)	Ropivacaine 0.5%, 40 mL bolus [67] (Low)	Ropivacaine 0.2% (Low)	5 mL/h (Low)	48 h (Low)
	Close to the bone (Moderate) Multi-orifice (Moderate)	Ropivacaine 0.3%, 20 mL bolus [68] (Low)	Bupivacaine 0.5% Ropivacaine 0.3–0.5% (Low)	8–10 mL/h (Low)	60–72 h (Moderate)
Open major digestive tract surgery (colorectal) [4]	Preperitoneal space Cephalad catheter orientation (Moderate) Multi-orifice (Moderate)	Ropivacaine 0.2%, 10 mL (Moderate)	Ropivacaine 0.2% Bupivacaine 0.25% Levobupivacaine 0.25% (Moderate)	10 mL/h (Moderate) or intermittent bolus 8–10 mL repeated at 5 to 12 h (Moderate)	48 h (Moderate)
Open hepatobiliary surgery (subcostal incision) [4]	Preperitoneal or in a musculo-fascial layer (Moderate) Multi-orifice (Moderate)	Bupivacaine 0.5%, 10 mL Ropivacaine 0.25%, 20 mL (Low)	Ropivacaine 0.25% Bupivacaine 0.5% (Moderate)	4 mL/h (High) intermittent bolus 10 mL repeated at 4 or 12 h (Moderate)	At least 48 h (Moderate)
Laparoscopic cholecystectomy [4]	Gall bladder bed and trocar sites (Low) Epidural/Multi-orifice (Moderate)	Ropivacaine 0.5%, 20 mL Intraperitoneally and at trocar sites [69] (Low)	Ropivacaine 0.5% (Low)	Incremental doses of 10 mL (Low)	Not given
Open appendectomy [4]	Preperitoneal (Moderate) Epidural (Low)	Ropivacaine 0.2%, 10 mL [70] (Moderate)	Ropivacaine 0.2% (Low)	5 mL/h (Moderate)	24 h (Low)
Nephrectomy [4]	Between transverse and oblique intern muscles (Low) Epidural/Multi-orifice (Low)	Bupivacaine 0.25%, 20 mL Ropivacaine 1%, 10 mL (low)	Bupivacaine 0.25% Ropivacaine 0.5% (Very low)	At least 4 mL/h (Very low)	48 h (Very low)
Inguinal herniotomy [4]	Subfascial (Moderate) Epidural/Multi-orifice (Low)	Bupivacaine 0.25%, 20 mL Bupivacaine 0.5%, 10 mL (High)	Bupivacaine 0.5% (Low)	2 mL/h (Low)	48 h (Moderate)
Cesarean section [4]	Above/below fascia (Moderate) Multi-orifice (Moderate)	Bupivacaine 0.125–0.25%, 25 mL [71] Ropivacaine 0.2% Levobupivacaine 0.125% (Moderate)	Bupivacaine 0.125–0.25%, 25 mLRopivacaine 0.2% Levobupivacaine 0.125% (Moderate)	5 mL/h -	72 h (Moderate)
Abdominal hysterectomy with bilateral salpingo-oophorectomy [4]	Above fascia (Moderate) Multi-orifice (Moderate)	Levobupivacaine 0.25%, 20 mL [72] (Moderate)	Bupivacaine 0.25–0.5% Ropivacaine 0.1–0.2% Levobupivacaine 0.25% (Moderate)	5 mL/h [72] -	52 h (Moderate)
Retropubic prostatectomy [4]	Subfascial (beneath rectus muscle) (Very low) Multi-orifice (Very low)	-	Ropivacaine 0.2% Bupivacaine 0.5% (Very low)	5 mL/h (Very low)	48 h -
Median sternotomy [4]	Two catheters in different wound layers (subfascial plane and subcutaneous) (Moderate) Multi-orifice (Moderate)	-	Ropivacaine 0.2% (Moderate)	At least 4 mL/h (2 mL/h for each catheter) or intermittent boli 5 mL/h per catheter (Low)	48 h (Moderate)
Thoracotomy [4]	Subcutaneous (Low) Epidural (Low)	-	Ropivacaine 0.2% (Low)	At least 2 mL/h (Low)	48 h (Low)
Breast surgery [4] Modified radical mastectomy and axillary node dissection Elective cosmetic breast augmentation Bilateral breast augmentation	Axillary wound cavity (Moderate) Multi-orifice (Moderate)	-	Bupivacaine 0.5% (Moderate)	0.5 mL/h (Moderate)	5 days (Moderate)
	Subcutaneous (Moderate) Multi-orifice (Moderate)	Suggested not evaluated	Ropivacaine 0.25% (Moderate)	Intermittent 10 mL on demand [73] (Moderate)	48 h (Moderate)
	Catheter tip superior to the prothesis (Moderate) Multi-orifice (Moderate)	Suggested not evaluated	Bupivacaine 0.25% (Moderate)	2 mL/h at catheter for each breast [74] (Moderate)	48 h (Moderate)

WI-wound infiltration. * Here we present the duration of infusion based on experts’ opinion with the note that CWI duration “should be tailored to the patient’s needs”.

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
