# Peer review of "Updates on Wound Infiltration Use for Postoperative Pain Management: A Narrative Review"

_jcm, 2021, doi:10.3390/jcm10204659_

Round 1

Reviewer 1 Report

the authors undertook an exhaustive narrative review of local anesthetic single shot wound infiltration and continuous wound infiltration for different surgical sites with some comparisons to neuraxial and peripheral nerve blocks. generally speaking they were very thorough with their comparisons and i only have a few requests for their consideration.

  1. systemic lidocaine administration (intravenous lidocaine infusion) has also been evaluated favorably in a number of surgical models, however may not be an option when added to wound infiltration with large volumes of local anesthetics. could the authors care to add a paragraph or two comparing or discussing this modality?
  2. throughout the MS, when discussing dose calculations for local anesthetics, the authors use body weight, not adjusted, not ideal. since the incidence of obesity is unfortunately high, a safer recommendation should be to use ideal body weight or at the very least great caution with obese individuals. 

minor comments: 

please check the use of " " on pg 3 ln 88-92. some are not correct, being at the bottom of the line, inappropriate spaces.

please add true local anesthetic allergy to pg 3 ln 104-5 as a contraindication.

figure 2 on pg 4 is very professional. just to be sure, it is the authors' original work or do they need permission to use?

may consider moving #4. local anesthetics and medications for wound infiltration to before #3 wound infiltration technique because the clinician has to have the drug selected before preforming the technique.

pg 5 ln 176, may consider the word pleotropic rather than "alternative"

pg 5 ln 206, when discussing WI, motor block is not pertinent like it would be in plexus or neuraxial blocks. 

pg 6 ln 220, i recommend stating dose local to ideal body weight, not total body weight.

pg 12 ln 401, please consider rewording to, "... in one study CWI showed efficacy comparable to epidural analgesia for ___ (specific surgical stimulus)"

pg 12 ln 402-404, similar to above, please add specific surgical stimulus. 

pg 20 ln 795, "Use liposome vesicles . . . minimize side effects." awkward sentence. i believe they meant use of liposomal bupivacaine

pg 20 ln 808-809, peripheral nerve blockade at compressible sites is NOT contraindicated in patients with coagulation abnormalities or on anticoagulants

pg 21 ln 834, what is EFIC PAIN OUT?

pg 21 ln 840, minor error in use of ""

figure 3: doses of local anesthetic should probably be calculated on ideal body weight due to the prevalence of obesity

Author Response

REVIEWER #1:

1

Reviewer 1

Comments and Suggestions for Authors

the authors undertook an exhaustive narrative review of local anesthetic single shot wound infiltration and continuous wound infiltration for different surgical sites with some comparisons to neuraxial and peripheral nerve blocks. generally speaking they were very thorough with their comparisons and i only have a few requests for their consideration.

1)systemic lidocaine administration (intravenous lidocaine infusion) has also been evaluated favorably in a number of surgical models, however may not be an option when added to wound infiltration with large volumes of local anesthetics. could the authors care to add a paragraph or two comparing or discussing this modality?

In response to reviewer’s comment, we included the following paragraph in section 8. Future directions (page 22, line 903-912):

ERAS protocols strongly recommend intravenous lidocaine infusion for open and laparoscopic colorectal surgery [262], due to opioid-sparing effect, with reduced incidence of nausea, vomiting [273] and postoperative ileus [274]. Published data suggest that toxicity of intravenous lidocaine is related to plasma concentration, and although toxicity is infrequent, monitoring patients for signs of toxicity is mandatory [275]. Since lidocaine infusion has been used in genitourinary, gynecology, ambulatory, breast,  spine and cardiothoracic surgery [274], the idea of combining iv lidocaine and WI with LA sounds tempting. However, because the pharmacokinetic profile of combining IV lidocaine infusion with LA WI has not been studied, safety is a major concern as it is reasonable to expect higher incidence of toxic effects.

2)throughout the MS, when discussing dose calculations for local anesthetics, the authors use body weight, not adjusted, not ideal. since the incidence of obesity is unfortunately high, a safer recommendation should be to use ideal body weight or at the very least great caution with obese individuals. 

In response to the above reviewer comment the corrections were performed throughout the manuscript where the term was used (answers are given in detail in the next section answers #9 and #16).

minor comments: 

3) please check the use of " " on pg 3 ln 88-92. some are not correct, being at the bottom of the line, inappropriate spaces.

In response to the above reviewer comment we edited the manuscript as follows (page 3, line 91-92): We searched PubMed for abstracts in English, using the terms “wound infiltration AND postoperative pain AND”, “neurosurgery”, “cardiac surgery”, “trauma surgery”, “emergency cases”, “thoracic surgery”, “abdominal surgery”, “breast surgery”, “thyroid surgery”, ”day case surgery”, ”urology surgery”, ”gynecology surgery”, “othopedics”, “wound infection“, “wound bleeding“…

4) please add true local anesthetic allergy to pg 3 ln 104-5 as a contraindication.

In response to the above suggestion we edited the manuscript as follows (page 3, line 104-5): Infection at the site of injection, the true LA allergy and patient refusal are the only contraindication for WI [15].

5) figure 2 on pg 4 is very professional. just to be sure, it is the authors' original work or do they need permission to use?

In response: Figure 2 is the work of Ilija Ilic, a graphic designer whose name is listed in acknowledgment, based original drawings by the author Mihailo Bezmarevic, MD.

6) may consider moving #4. local anesthetics and medications for wound infiltration to before #3 wound infiltration technique because the clinician has to have the drug selected before preforming the technique.

We hope that the reviewer will respect our decision to leave the order of sections #3 and #4 as it is.

7)pg 5 ln 176, may consider the word pleotropic rather than "alternative"

In response: the word “pleotropic” does not exist in the Merriam-Webster dictionary. However, the word “pleiotropic”, which is defined as “producing more than one effect”, is more accurate, compared to the word “alternative” as it is used in line 176. Therefore, based on the above reviewer suggestion, we entered the word “pleiotropic” instead of the word “alternative” in page 5, line 176 of the revised manuscript we are now submitting for your consideration.

8)pg 5 ln 206, when discussing WI, motor block is not pertinent like it would be in plexus or neuraxial blocks. 

In response to the above suggestion we edited the manuscript as follows (page 6, line 206): Adding epinephrine extends anesthesia duration and motor blockade, but when discussing WI, motor block is not a pertinent consideration [16,45].

9)pg 6 ln 220, i recommend stating dose local to ideal body weight, not total body weight.

In response to reviewer’s comment the corrections are performed throughout the manuscript (page 6, line 221):

It is vital to limit the LA dose based on patient ideal body weight (IBW) [56] and risk factors (age, lower muscle mass, lower ejection fraction, liver and renal insufficiency, and metabolic disorders) [55].

10) pg 12 ln 401, please consider rewording to, "... in one study CWI showed efficacy comparable to epidural analgesia for ___ (specific surgical stimulus)"

In response to the above suggestion, we edited the text as follows: Meta-analysis presented CWI efficacy comparable to epidural analgesia for different incision types like subcostal, midline or transverse incisions [87].

11)pg 12 ln 402-404, similar to above, please add specific surgical stimulus. 

In response to the above suggestion we edited manuscript as follows: Recovery parameters, opioid consumption, associated side effects and patient satisfaction seemed to be in favor of preperitoneal wound catheters compared to epidural analgesia for midline incisions and L-shaped incisions [26,30].

12)pg 20 ln 795, "Use liposome vesicles . . . minimize side effects." awkward sentence. i believe they meant use of liposomal bupivacaine

In response: Based on the above comment, we corrected the manuscript as follows (page 20, line 798): Use of liposomal structures not only for bupivacaine, but also for NSAIDs decreases inflammation after local injection, improves NSAIDs' effectiveness and minimizes side effects [247].

13)pg 20 ln 808-809, peripheral nerve blockade at compressible sites is NOT contraindicated in patients with coagulation abnormalities or on anticoagulants

In response to the above comment we edited the manuscript as follows (page 20, line 814): Local infiltration analgesia, WI and CWI are viable alternatives when peripheral nerve blocks cannot be performed due to lack of staff or equipment [251], when motor block is undesirable and there is need for immediate mobilization [5,243], and in patients with coagulation abnormalities or on anticoagulation therapy (with exemption of compressible sites where peripheral nerve blocks are not contraindicated) [3,252].

14)pg 21 ln 834, what is EFIC PAIN OUT?

In response to the above question, we revised the manuscript as follows (page 21, line 898): Based on unpublished experience from the European Pain Federation (EFIC) supported international quality improvement project under the name PAIN OUT, we propose implementation of WI as part of multimodal management whenever possible, particularly when other regional anesthesia techniques are contraindicated.

15)pg 21 ln 840, minor error in use of ""

In response to the above suggestion we edited the text as follows (page 22, line 914):…use “one size fits all” LA

16)figure 3: doses of local anesthetic should probably be calculated on ideal body weight due to the prevalence of obesity

In response to the above suggestion, we edited figure 3.

Reviewer 2 Report

This is a valuable review about updates on wound infiltration use for postoperative pain management. The manuscript is well organized and described. However, I have some suggestions as following:

  1. Since wound infiltration (WI) is a part of multimodal analgesia for postoperative pain control and enhanced recovery after surgery (ERAS). Please discuss the role of WI in ERAS in detail.
  2. Please describe the use of WI in not only laparoscopic cholecystectomy but also other laparoscopic surgeries because laparoscopic surgery has been popular for minimally invasive surgery.

Author Response

Reviewer 2

Comments and Suggestions for Authors

This is a valuable review about updates on wound infiltration use for postoperative pain management. The manuscript is well organized and described. However, I have some suggestions as following:

1.Since wound infiltration (WI) is a part of multimodal analgesia for postoperative pain control and enhanced recovery after surgery (ERAS). Please discuss the role of WI in ERAS in detail.

In response to the above comment, we edited the manuscript as follows (page 21, line 832-888):

  1. Wound infiltration in enhanced recovery after surgery protocols

Enhanced recovery after surgery (ERAS) is the gold standard in contemporary surgical practice. ERAS protocols aim to reduce the perioperative stress response in order to accelerate patient recovery and return to daily activities. Use of multimodal analgesia is an important part of ERAS protocols, in an attempt to reduce or even eliminate opioid use and therefore promote early mobilization and bowel motility, prevent nausea and vomiting, and avoid the long-term consequences of opioid use [256]. Thus, regional analgesic techniques that include neuraxial (e.g., epidural, spinal), peripheral nerve blocks, and wound infiltration are part of current ERAS protocols.

Recent guidelines for enhanced recovery after lung surgery suggest multimodal analgesia, including regional analgesia or local anesthetic techniques, in an attempt to avoid or minimize opioids and their side effects [113]. ERAS protocol updates need to promote the use of WI in VATS, where current evidence suggests that WI is very effective [113]. Guidelines for ERAS after cardiac surgery do not include WI [257], but further research is needed in this field. Similarly, esophageal surgery ERAS protocols do not mention WI as an analgesic option [258], whereas the ERAS Society recommends WI with LA particularly with ropivacaine or levobupivacaine [259] after bariatric surgery (high evidence level, strong grade of recommendation). In addition, pre-incision WI [260] combined with intraoperative bupivacaine aerosolization [261] may present a reasonable option for enhancing recovery after bariatric surgery [259]. Although there are no clear recommendations about safe doses of LAs in bariatric surgery ERAS protocols, doses of local anesthetic should be calculated based on patient’s ideal body weight (IBW), in order to reduce the risk of LA toxicity.

Although published studies support the use of CWI or WI in open colorectal surgery, current ERAS protocols do not recommend its use [262]. ERAS recommendation for rectal/pelvic surgery states that there is low evidence level and therefore weak recommendation for CWI via pre-peritoneal catheters due to “limited evidence” from ERAS protocol-based studies [263]. However, there is clear recommendation for CWI through preperitoneal catheter as “alterantive to epidural” in ERAS for open pancreaticoduodenectomy (high evidence level, strong grade of recommendation) [264]. 

ERAS protocol for major head and neck cancer surgery with free flap reconstruction recommends only systemic analgesia [265]. In neurosurgery, although scalp infiltration and scalp blocks can be recommended for craniotomies, there is no ERAS Society protocol due to lack of evidence [266,267].

ERAS protocols in urology recommend epidural analgesia for open abdominal and pelvic procedures [268]. However, available data suggest the use of CWI with pre-peritoneal catheters combined with systemic analgesia (paracetamol and NSAIDs) for minimally invasive surgical procedures instead of different types of regional analgesia and intravenous lidocaine [268]. The recent update of ERAS for gynecological procedures recommends WI with bupivacaine (high evidence level) while noting that studies are needed to compare thoracic epidural analgesia vs. transversus abdominis block and WI [269].

In orthopedic surgery, ERAS recommends LIA with LA for knee replacement (evidence level high, recommendation grade strong), but not for hip replacement [270]. The authors explain that the advantages of LIA over peripheral nerve blocks and neuraxial blocks include the absence of motor blockade, thus enabling early mobilization, the preservation of hemodynamic stability and the absence of influence on urine retention [270]. In vascular surgery, a recent systematic review suggested that use of ERAS protocols is currently limited because of low quality evidence regarding feasibility and effectiveness [271]. One RCT comparing thoracic epidural analgesia vs. CWI analgesia after abdominal aortic surgery showed that CWI resulted in comparable, good pain control but required higher doses of LA [272]. Because there is potential for WI or CWI to be beneficial, additional high-quality studies are needed to evaluate WI and CWI as part of multimodal recovery after vascular surgery [271]. We believe that WI can be an important addition to ERAS protocols, and has the potential to be a low cost, easy and safe alternative to other techniques currently in ERAS protocols, such as nerve blocks or neuraxial analgesia.

2.Please describe the use of WI in not only laparoscopic cholecystectomy but also other laparoscopic surgeries because laparoscopic surgery has been popular for minimally invasive surgery.

In response to reviewer’s comment: Studies refering WI in laparoscopic surgery have been carefully searched and cited throughout the manuscript in esophagogastric surgery, hepatic surgery, colorectal surgery, gynecological surgery.

P 14.Geriatric patients undergoing laparoscopic gastrectomy who received single shot WI with bupivacaine (0.5%, 40 ml) had lower postoperative pain scores and lower morphine consumption for 48 hours compared to placebo [153].

P15. In patients undergoing laparoscopic hepatectomy, WI and ropivacaine infused gelatin sponge placed on the liver cutting surface provided lower pain scores at rest and on movement, reduced opioid consumption, and lower stress hormones levels during 48 hours compared with placebo [163].

P15.In patients undergoing laparoscopic colon resection, CWI ropivacaine combined with systemic ketorolac and propacetamol after surgery showed similar efficacy, postoperative inflammatory response, incidence of wound-related complications, and cancer recurrence in comparison to PCA-IV opioid during 48 hours [173]. No difference in CWI efficacy was observed between ropivacaine and lidocaine for 48 hours [174]. Single-shot WI with bupivacaine at the end of laparoscopic single-incision colectomy resulted in lower pain scores and lower analgesic consumption compared to no intervention [175].

P19. CWI with ropivacaine provided similar analgesic effects as PCA fentanyl and ketorolac after laparoscopic gynecologic surgery, and despite higher rescue analgesic use, benefits included opioid-sparing effects and fewer side effects during 24 hours follow up [228]. Single WI with levobupivacaine [224,225], bupivacaine [226], or liposomal bupivacaine [227], in addition to general anesthesia and standard analgesic therapy including NSAIDs or paracetamol and opioids significantly decreased postoperative analgesic requirement [224-227], lowered pain intensity [224,225,227], and reduced time to ambulation after laparoscopic [224,225,227], and open gynecological surgery [226].

Pre-incision port site infiltration with liposomal bupivacaine compared with bupivacaine decreased pain on the second and third postoperative day after laparoscopic or robotic multiport hysterectomy [227]. Surgical approach may influence postoperative pain when WI is used, as patients needed less opioid after laparoscopic gynecological surgery compared to transabdominal surgeries [233].

This manuscript is a resubmission of an earlier submission. The following is a list of the peer review reports and author responses from that submission.

Round 1

Reviewer 1 Report

It is of high interest for the clinicians (both surgeons and anaesthesiologists) to have an updated review on the status of local anaesthesia infiltration for post-operative pain relief. Stamenkovic and co-workers have done an extensive and impressive work in collecting 227 relevant references on the subject, and provide some general aspects of local anaesthesia infiltration as well as adressing individual studies in a procedure specific sequence of paragraphs.

  • While they describe the exclusion of children and plastic surgery (which are OK) and also a proper selection of key-words, they do not say how long back in time they found appropriate papers, and, more importantly, they do not say anything about the criteria for include some papers whereas others are not included.
  • They define and distinguish between the terms «wound infiltration» (WI), «continuous wound infiltration» (CWI) and «local infiltration anesthesia» (LIA). The terms LIA and WI are used somewhat inconsistently through the manuscript, and their term LIA is problematic for two reasons: 1) It is usually a question of analgesia, not anesthesia in the studies 2) The term LIA is usually used on the technique of extensive infiltration of high volume, low concentration of local anesthetics done by the surgeon in the wound, typically used for total knee end hip arthroplasty (but also others)
  • Another important distinction, which should be made, is between local anesthesia infiltration and nerve blocks. This should be addressed better: When is a nerve block better? When is a nerve block not feasible?
  • The authors write a lot on toxicity, without really getting down to practical recommendations: How much for which type of procedure? What about total toxicity when mixing drugs? The mandatory need for intralipid at hand (not mentioned?)
  • The individual studies are usually reported on being effective (for the local anesthesia part) or not, in some cases the comparator or control group is mentioned.  This should be more systematically approached for each study: What was the comparator?  (placebo or active control), What did the study groups receive of other non-opioid analgesics? What was the extent and duration of the positive effect.
  • The authors have some statements on preoperative infiltration being better than by end of surgery, without showing good documentation of this interesting point.
  • The authors have a lot of references to studies in which local anesthetics have been mixed successfully with adjuvants, such as ketamin, NSAID etc, without checking/stating if the study in question have corrected for the systemic effect of such adjuvants. For this purpose the adjuvant should be tested against the same dose/drug given iv in order to see if the local effect is of importance.
  • The tables are not much informative, only containing a selection (on which criteria?) of studies with their type of surgery, local anesthestic (type and dose), single or continuous and timing. If these tables should be included they should contain information on comparator group and a few words on effect strength and duration.

Some minor comments:

  1. Line 63: catheters are also limited by displacements and infection risk
  2. Line 154: you should discuss the physiology and benefits with bolus refill of catheter versus an infusion
  3. Line 148: some reference on pre-operative block being better=
  4. Line 196: usually adrenaline enforce the motor block by keeping more drug near the nerve for a longer time
  5. Line 700: It is not correct to say that liposomal bupi offers excellent analgesia for 72 hrs. The extra effect is rather limited.